# Molecular Mechanisms of Malignant Transformation of Oral Submucous Fibrosis by Different Betel Quid Constituents—Does Fibroblast Senescence Play a Role?

**DOI:** 10.3390/ijms23031637

**Published:** 2022-01-31

**Authors:** Pangzhen Zhang, Nathaniel Quan En Chua, Simon Dang, Ashleigh Davis, Kah Wee Chong, Stephen S. Prime, Nicola Cirillo

**Affiliations:** 1School of Agriculture and Food, Faculty of Veterinary and Agricultural Sciences, The University of Melbourne, Parkville, VIC 3052, Australia; pangzhen.zhang@unimelb.edu.au; 2Melbourne Dental School, Faculty of Medicine, Dentistry and Health Sciences, The University of Melbourne, Carlton, VIC 3053, Australia; chua5@student.unimelb.edu.au (N.Q.E.C.); simon.dang@student.unimelb.edu.au (S.D.); davisaj@student.unimelb.edu.au (A.D.); kahc2@student.unimelb.edu.au (K.W.C.); 3Centre for Immunology and Regenerative Medicine, Institute of Dentistry, Barts and the London School of Medicine and Dentistry, Queen Mary University of London, London E1 4NS, UK; stephensprime@gmail.com

**Keywords:** betel quid, areca nut, arecoline, fibroblast senescence, oral carcinogenesis, oral submucous fibrosis, carcinogenic potential

## Abstract

Betel quid (BQ) is a package of mixed constituents that is chewed by more than 600 million people worldwide, particularly in Asia. The formulation of BQ depends on a variety of factors but typically includes areca nut, betel leaf, and slaked lime and may or may not contain tobacco. BQ chewing is strongly associated with the development of potentially malignant and malignant diseases of the mouth such as oral submucous fibrosis (OSMF) and oral squamous cell carcinoma (OSCC), respectively. We have shown recently that the constituents of BQ vary geographically and that the capacity to induce disease reflects the distinct chemical composition of the BQ. In this review, we examined the diverse chemical constituents of BQ and their putative role in oral carcinogenesis. Four major areca alkaloids—arecoline, arecaidine, guvacoline and guvacine—together with the polyphenols, were identified as being potentially involved in oral carcinogenesis. Further, we propose that fibroblast senescence, which is induced by certain BQ components, may be a key driver of tumour progression in OSMF and OSCC. Our study emphasizes that the characterization of the detrimental or protective effects of specific BQ ingredients may facilitate the development of targeted BQ formulations to prevent and/or treat potentially malignant oral disorders and oral cancer in BQ users.

## 1. Introduction

Betel quid (BQ) chewing is a common habit in many parts of Asia and some Pacific Islands; approximately 600 million people worldwide chew some form of BQ regularly [1,2]. BQ chewing is typically associated with malignant and potentially malignant conditions of the oral cavity, including oral squamous cell carcinoma (OSCC) and oral submucous fibrosis (OSMF), respectively [3]. OSMF is a chronic, potentially debilitating condition that is characterized pathologically by juxta-epithelial inflammation, fibro-elastic changes in the lamina propria and epithelial atrophy, all of which clinically are associated with progressive rigidity of the oral mucosa. The disorder commonly affects any part of the oral cavity and sometimes can include the pharynx. OSMF is a major global health issue because of its high malignant transformation rate (up to 15–30% according to different studies) [4,5] and subsequent high mortality due to OSCC [5]. The historical background, etiological factors, pathogenesis, clinical features, differential diagnosis and management of this condition have been reviewed recently [6]. What is not yet known, however, is the function of the different BQ constituents in malignant transformation.

Different BQ ingredients have distinct effects in cells, tissues and living organisms. The formulation of BQ depends on a variety of factors that may include the cultural background of a country, regional backdrop, personal likings and availability of specific ingredients [1,2]. BQ is typically made up of areca nut, betel leaf, slaked lime and may or may not contain tobacco. Other substances such as mustards, sweeteners, cloves, saffron and cardamom may also be added [1]. In parts of Taiwan and Indonesia, inflorescence/flower is added to the quid for its aromatic flavor and in India, a variety of spices may be added [7].

We believe that it is important to gain more insight into the carcinogenicity of BQ ingredients. Very little is known, for example, about whether there is a correlation between the specific chemical components of BQ and the potential for malignant transformation in the oral cavity. To understand the role of different BQ ingredients in the pathogenesis of fibrosis and cancer, as well as to shed light on the mechanisms by which cellular senescence promotes OSMF and its malignant transformation, a structured search strategy was developed. Eligible articles indexed in MEDLINE/PubMed, Scopus, and Web of Science (WoS) were included. The search strategy used in this narrative review combined the search terms related to three categories, namely: BQ composition (1), malignant transformation of OSMF (2), and cellular senescence in OSMF and OSCC (3). The following search terms were utilized for the search: ‘(betel or masala or (stem and quid) or (areca and nut) or betel quid or (slaked and lime) or (betel and leaf) or catechu or (betel and inflorescence) plus (1) (analys* or compos* or alkaloid* or chemical* or ingredient*) not (lesion or effect or disorder or tea or grapevine or wine or lipase or apple)’ or (2) ‘(oral carcinogenesis) or (oral cancer) or (oral submucous fibrosis) or (malignant transformation)’ or (3) ‘(cellular senescence) or (fibroblast senescence) or (Senescence-Associated Secretory Phenotype) and (oral cancer) or (oral submucous fibrosis)’. The articles screened from this search (in the English language only) were used as the basis for the elaboration of the manuscript. Further grey literature was searched as appropriate.

This review aimed to survey the major chemical constituents of BQ and to offer a qualitative assessment of their potential role in oral carcinogenesis. The findings may improve our understanding of the pathophysiology of BQ-associated OSCC and OSMF and may guide future chemopreventive and therapeutic strategies.

## 2. Why It Is Important to Understand the Effects of Individual BQ Ingredients

Different types of BQ contain diverse chemical constituents. Areca nut, also known as betel nut, is the seed of the areca palm (*Areca catechu*) tree, typically grown in tropical regions. Alkaloids are one of the many well-known chemical compounds associated with areca nut [8], but other major chemical constituents have been known for decades and include crude fibre, proteins, lipids, carbohydrates, tannins, flavonoids and minerals [9]. Importantly, the concentration of each chemical constituent varies between the raw and ripe nut. Betel leaf (*Piper betle* L.) contains betel oil, which consists of phenolic compounds such as eugenol, chavicol and hydroxychavicol [10]. Slaked lime (calcium hydroxide) is also a popular additive and facilitates the entry of alkaloid stimulants into the bloodstream via sublingual absorption [11].

Areca nut chewing was once believed to have several health benefits such as improving digestion, psychomotor stimulation and anthelmintic activities [12]. Other potentially beneficial actions of BQ chewing include relaxation, improved concentration and mood enhancement. Truck drivers, for example, chew BQ to stay awake during their long hours of work [13]. More recent and conclusive evidence, however, indicates that the consumption of areca nut alone, or in the form of BQ, has negative health effects [14]. In particular, BQ, with or without tobacco, is carcinogenic in humans and its consumption has been associated with OSMF and OSCC, together with other cancers [14]. Chronic use of BQ has also been associated with chemical dependency and substance abuse [15] and long-term use is believed to increase the risk of heart disease [16], chronic kidney disease [17] and liver cirrhosis [18]. With specific reference to the mutagenic effects of extracts of BQ, the effect of areca nut on cultured cells has been studied extensively [19]. Nitrosation reactions involving areca nut alkaloids are thought to give rise to a variety of BQ-specific formation of nitrosamines. These, in turn, induce and propagate oxidative stress by interacting with DNA, proteins and other macromolecules. Together, they contribute to carcinogenesis in the oral mucosa, both in the epithelium and sub-mucosal connective tissues, a process that involves EGF/EGFR, IL-1α, ADAMs, JAK, Src, MEK/ERK, CYP1A1 and COX signalling pathways [20,21].

However, areca nut is almost always consumed in a BQ package that contains additional components such as betel leaf or betel stem inflorescence and slaked lime, both of which can affect the carcinogenic potential of the quid [8,22]. In this regard, we have shown recently that the composition of BQ varies geographically and this may explain the different levels of pathogenicity detected in vivo [23]. Hence, understanding the composition and preparation of BQ mixtures is key to the study of their clinical and molecular effects.

## 3. Chemical Constituents of Betel Quids

Several chemical constituents of BQ have been investigated in original studies [24,25,26,27,28,29,30,31,32], with arecoline being consistently identified in all samples (Table 1). Regardless of the product type, regional location and maturation of the nut, alkaloids have been consistently reported as the key drivers of OSMF and OSCC in BQ chewers. Whilst these compounds are the most extensively studied and have been shown to have strong biological activity, other compounds found in BQ, such as betel leaf, may also be important for the pathophysiology of BQ-related diseases.

### 3.1. Alkaloid: Arecoline

More than half a century ago, arecoline was reported to be the primary alkaloid in BQ [33]. Since that time, it has been detected in both areca nut products and in the areca nut itself. Saliva samples of BQ chewers have shown peaks of arecoline at *m*/*z* 156.1 [29], and, in 11 different pan masala blends, arecoline is present at 132–415 ng/mg in all the dry samples. High concentration of arecoline can be found in both young and mature BQ extracts [25]. Srimany and colleagues [27] supported these findings but showed that arecoline concentration varied between different sites in the nut. At Stage 2 of maturation, arecoline, arecaidine, and guvacoline were detected in the white region of the nut, whereas at Stage 3 of maturation, higher amounts of each constituent were detected in the brown region of the nut [27].

Arecoline concentration also varies geographically. In our recent study investigating the chemical composition of Indonesian BQ mixtures [8], the highest arecoline concentration was observed in the nuts and BQ mixtures from West Papua; BQ mixtures from other regions of the country showed a considerable reduction of arecoline concentration.

Taken together, these findings show that arecoline is abundant in areca nut and its products. The amount of bioavailable arecoline, however, might be modified by other components of the BQ chewing package and this aspect warrants further investigation.

### 3.2. Alkaloid: Guavacine

Guvacine is another prominent constituent of areca nut and BQ. It has been reported that guvacaine is the most abundant of the four main alkaloids (1.39−8.16 mg/g), followed by arecoline (0.64−2.22 mg/g), arecaidine (0.14−1.70 mg/g) and guvacoline (0.17−0.99 mg/g) [24]. In the same study, saliva from pan masala and gutkha users had higher levels of guvacine (*p* < 0.001), whilst levels of the other alkaloids were unchanged [24]. Interestingly, levels of guvacine in areca nut are also influenced by the maturation period of the nut. Specifically, guvacine levels were six times higher in the mature versus young betel nut extracts (1871.4 ug/g vs. 302.9 ug/g) [25]. These findings are consistent with the results of Srimany and colleagues [27] who reported that the white region of a ripened nut contained only guvacine and no other alkaloid.

Whilst these studies are important and document the presence of guvacine in BQ, there is no clear data regarding how guvacine relates to oral carcinogenesis.

### 3.3. Alkaloid: Guvacoline and Arecaidine

In different areca nut products, Jain et al. [24] identified trace levels of arecaidine and guvacoline (0.14–1.70 mg/g dry weight), levels that were lower than arecoline (0.64–2.22 mg/g) and guvacine (0.17–0.99 mg/g). Similarly, arecaidine and guvacoline have been identified in saliva following chewing of three BQ preparations [25]. In a different study, only a small amount of guvacoline was detected in a pan masala blend compared to arecoline and nicotine (arecoline 7.341 mg/g; guvacoline 7.092 mg/g; nicotine 7.480 mg/g); no mention of arecaidine was made in this study [31]. Interestingly, mass spectrometric data have shown that the ripened areca nut contains arecoline, arecaidine and guvacoline in its brown region and, as the nut matures, these alkaloids preferentially stay in that region, albeit at lower levels [27]. In cell transformation assays, arecoline has significant mutagenic potential that exceeds that shown by arecaidine [28]. Pan masala has also been examined due to its acknowledged carcinogenicity, but most studies have focused on arecoline only rather than guvacoline and arecaidine [32].

### 3.4. New Alkaloids and Additives

New alkaloids have been identified in Areca catechu including Acatechu A and B [26] and Arecatemines A–C [30]. These studies focused on the isolation of these new alkaloids and did not report their association with oral carcinogenesis.

Additives are popular in many BQ preparations worldwide. Kadi et al. [29] reported 156 *m*/*z* arecoline in mass spectra of the saliva of BQ chewers with and without the addition of slaked lime. Tobacco is another popular additive and, in the study by Lord et al. [31], nicotine was detected through CZE analysis and had a peak of 7.480 m/volts. This is consistent with the findings of Franke and colleagues [25] who demonstrated that nicotine was the main alkaloid in tobacco extracts (15,454.8 μg/g), with relatively negligible amounts detected in the nut and leaf extracts (0.1–1.4 μg/g). The role of nicotine and smokeless tobacco in oral carcinogenesis has been studied extensively [2] and is not the focus of this review.

### 3.5. Polyphenols

Polyphenols are common constituents of BQ ingredients including areca nut and leaf, husk and stem inflorescence (SI). Phenolic acids have been detected mostly in the husk, leaf and SI, but traces have also been identified in areca nuts from BQ mixtures. In a recent study, flavanols (predominantly catechin and epicatechin) were the most abundant polyphenols detected in Indonesian BQ and concentrations appeared to be higher in nuts than leaf, husk and SI [8]. Flavonoids and tannins have also been identified [26].

### 3.6. Other Chemical Consituents

Chavibetol is an allyl benzene found in *Piper betle* leaves [25]. It has been detected in saliva samples of BQ chewers and appears to be absent in betel nuts. Whilst chavibetol is highly concentrated in leaf extracts (39,873.8 ug/g), the concentration range varies considerably and further evaluation is required.

Cao and colleagues [26] isolated and identified carbohydrates and fatty acids from dried *Areca catechu* L.; the compound structures in this study were determined by comparing their spectroscopic data with that reported in the literature. Other authors [27] also identified sugars and lipids in the maturing areca nut through localized NMR spectra.

## 4. Mechanisms of Malignant Transformation: Role of BQ Components

Smoking and alcohol are well-known risk factors of OSCC. However, malignant transformation of OSMF involves additional and potentially different pathways that are specific to OSMF. For example, it is established that the onset of fibrosis is critical in the malignant transformation of OSMF [34] but the mechanisms by which this process occurs are not been fully understood. Arecoline induces oxidative stress and cell cycle arrest in human keratinocytes [35] whereas in fibroblasts, the same oxidative damage activates pro-fibrotic transforming growth factor β1 (TGFβ1) [36]. In this regard, human studies have demonstrated elevated levels of oxidative stress biomarkers and their correlation with the level of fibrosis and clinical OSMF staging [37]. The irreversibility of fibrosis in response to BQ would then be the result of resistance exhibited by cross-linked collagen to proteinases [38]. Fibrosis in the connective tissue leads to a reduction in blood supply and the development of hypoxia. Importantly, the thickness of fibrosis in OSMF correlates with epithelial dysplasia [39].

Key triggers of subsequent malignant transformation of OSMF could be the inflammatory milieu and the production of reactive oxygen species (ROS). ROS is produced by both auto-oxidation of areca nut polyphenols in saliva and nitrosation of areca alkaloids. Following an arecoline-dependent increase in IL-6, IL-8 and GRO-α, repeated and continuous exposure to ROS has the potential to cause DNA double strand breaks and other pathogenic mechanisms of DNA damage in oral mucosal cells through oxidative stress [40]. IL-6, IL-8 and GRO-α can also promote the acquisition of the senescence-associated secretory phenotype (SASP) in fibroblasts which, in turn, induces an epithelial-to-mesenchymal transition (EMT) that facilitates invasion and metastasis in carcinomas [41]. Another contributor to malignant transformation of OSMF is the transcription factor hypoxia-induced factor 1α (HIF-1α). HIF-1α is upregulated during hypoxia and is responsible for activation of pro-angiogenic genes that encode vascular endothelial growth factor (VEGF) [42]. Thus, fibrosis and hypoxia appear to be critical in epithelial malignant transformation.

Below, we summarise the evidence supporting the pro- and anti-carcinogenic effects of individual constituents of BQ mixtures.

### 4.1. Areca Nut Alkaloids

Arecoline and guvacine are the best-studied compounds detected in areca nut and the relative proportions of arecoline and guvacine vary depending on the part of the areca nut that is sampled, as discussed previously. Early studies concluded that arecoline was the main alkaloid associated with the mutagenic potential of BQ because it induces chromosome breaks and causes other cellular aberrations in vivo [28]. Guvacine has also been implicated in oral carcinogenesis because its nitrosation leads to the formation of *N*-nitrosamine metabolites [43]. The four nitrosated derivates (*N*-nitrosoguvacoline, *N*-nitrosoguvacine, 3-(*N*-nitrosomethylamino)propionaldehyde and 3-(*N*-nitrosomethylamino) propionitrile) cause DNA single strand breaks and DNA protein cross-links that, in turn, cause a decrease in the survival of cultured human buccal epithelial cells. In addition, nitrosamines interact with other macromolecules of the cell through oxidative stress, thereby contributing to oral carcinogenesis [44,45,46]. Harvey and colleagues [47] originally found that arecoline and its hydrolysed product arecaidine stimulated collagen synthesis in cultured oral fibroblasts, which has been confirmed in more recent studies [48]. Interestingly, these arecoline-mediated pro-fibrotic effects may also be induced indirectly via oral keratinocytes, which, in turn, affect the collagen metabolism of fibroblasts [49]. Conversely, arecoline and its derivatives inhibit growth and collagen synthesis of oral fibroblasts when used at high concentrations, with 3-(*N*-nitrosomethylamino)propionaldehyde being the most cytotoxic and genotoxic derivative [50,51,52]. Arecoline has also been shown to induce inflammation and produce ROS in OSCC and OSMF [53,54,55].

In summary, analysis of the existing literature shows that areca nut constituents not only are associated with the formation of *N*-nitrosamine metabolites through nitrosation, the induction of ROS and the disruption of collagen breakdown causing increased collagen accumulation, but also are also involved in a variety of other processes such as ulceration from mechanical trauma of the coarse areca nut fibres, stimulation of fibroblast proliferation, activation of the coagulation system and cytotoxicity of oral epithelial cells. All of these factors are likely to contribute to the pathogenesis of OSMF and its malignant transformation [56].

### 4.2. Slaked Lime

The association between consumption of slaked lime and OSCC was proposed in seminal epidemiological research from Papua New Guinea [57]. The investigators found that oral cancer in this region is concentrated at the corner of the mouth and cheek, in striking contrast with western populations, and corresponds precisely with the site of application of lime in 77% of 169 cases. Subsequent studies have proposed that slaked lime is implicated in the development of inflammation and generation of ROS which, in turn, exerts harmful effects on DNA, proteins, lipids and other macromolecules [44,45]. In particular, the addition of slaked lime to areca nut creates an alkaline environment that potentiates the release ROS and is optimal for the production of hydrogen peroxide and superoxide radicals through auto-oxidation of both areca nut and polyphenols such as tannins and catechin [45].

Slaked lime also increases the proliferation of fibroblasts and causes phenotypic alterations in these cells [58], a phenomenon that has been attributed to the hydrolysis of arecoline to arecaidine. The phenotypically altered fibroblasts cause an increase in the production of collagen fibres which have an altered molecular structure [44]. Arecoline and arecaidine also work in concert at a transcriptional level and in fibroblasts, cause an increase in the deposition of collagen in the extracellular matrix by the production of large amounts of tissue inhibitors of metalloproteinases (TIMPs) [58]; these changes modulate cell proliferation, cell migration and invasion, as well as having anti- and pro-apoptosis activities [59]. Thus, slaked lime functions by altering the environmental conditions to favour the carcinogenic effects of BQ compounds.

### 4.3. Polyphenols and Other Compounds with Anti-Oxidant Properties

Five main polyphenols are commonly extracted from betel quid mixtures, namely catechins, flavonoids, flavan-3:4-diols, leucocyanidins and hexadroxyflavans [60]. In the main, polyphenols are regarded as anti-oxidants and, as such, these compounds may play a protective role in OSMF. However, their biological properties can be significantly modified by the chemical environment. For example, it is known that polyphenols are oxidised in the presence of slaked lime and cause the generation of ROS and the characteristic red hue of the saliva, teeth and lips of BQ users. Further, in the oral cavity the oxidative damage to DNA from ROS may be catalysed by trace copper and iron ions [61]; there are detectable copper levels in BQ mixtures and this trace element further increases collagen production and cross-linking by oral fibroblasts [51,62]. Interestingly, it has been shown that polyphenols can also act as pro-oxidants to induce apoptosis [63] which likely occurs as a protective mechanism. With regards to other compounds, in addition to the tannins (gallotanic acid in areca nut), catechins and flavonoids also increase fibrosis by disrupting collagen breakdown and promoting both collagen production and cross-linking [46]. This myriad of putative pre-neoplastic cellular alterations strongly emphasises the capacity of a variety of betel quid constituents to contribute to the pathogenesis of OMSF.

Research from Cirillo’s group has shown that the husk contains the widest range of polyphenols that are associated with high antioxidant capacity [8], suggesting that polyphenols at moderate levels are protective against diseases associated with oxidative stress (cancer, coronary heart disease, inflammation). This, in part, is due to the antioxidant activity of polyphenols through ROS scavenging, metal chelation and/or by the modulation of oxidative stress-related enzymes, all of which decrease DNA damage [63]. Petti and Scully [64] also alluded to the chemo-preventative role of polyphenols, particularly the capacity of methoxylated flavonoids to inhibit DNA adduct formation induced by carcinogens such as tobacco. These findings support several epidemiological studies that have linked flavonoids with high antioxidant activity [63].

In summary, the majority of the effects attributed to polyphenols are linked to their antioxidant and anti-inflammatory properties which, in turn, impact molecular events and signalling pathways associated with cell survival, proliferation, differentiation, migration, angiogenesis, detoxification enzymes and immune responses [65]. Unfortunately, polyphenols often have poor bioavailability when administered as pure active compounds, and their activity is modified in certain environmental conditions. For example, slaked lime alters the pH (~10) which damages polyphenols and reduces the antioxidant activity of these compounds [8], thus potentially promoting carcinogenesis [46]. Individuals who consume BQ with high concentrations of polyphenols, but also arecoline with slaked lime, are more likely to develop OSMF [23]. Interestingly, plant-derived polyphenols have both pro-oxidative and anti-oxidative properties depending on their metal-reducing potential, chelating behaviour, pH and solubility characteristics [66].

As such, the anti- or pro-oxidant potential of polyphenols depends greatly on the chemical nature, concentration and the micro-environmental conditions (cell types, pH, and redox stress) in which they are consumed [63]. These factors are particularly relevant to the pathophysiology of OSMF as the chemical interaction between different BQ ingredients and the oral environment determine the actual micro-environmental conditions. Furthermore, polyphenol concentration varies according to the geographic region in which the BQ is consumed and the degree of maturity of the areca nut itself. Together, these factors determine whether polyphenols have a protective or detrimental role in the oral cavity. Experiments are underway in our laboratory to investigate if and how these environmental conditions affect the anti-oxidant capacity of BQ polyphenols.

#### 4.3.1. Catechins

Catechins, which are abundant in the areca nut [8], have been shown to inhibit the production of metalloproteases and reduce invasion and migration [67]; concurrently, there is induction of apoptosis and growth arrest in OSCC and oral leukoplasia cell lines [68]. Along with safrole and hydroxychavicol, catechins also have the capacity of exerting cytotoxic activity in normal epithelial cells (Cirillo N, unpublished observations). Epigallocatechin gallate (EGCG), a major polyphenolic constituent of green tea, also has a significant, albeit transient (up to 48 h), ability to reduce the proliferation and migration of oral cancer cells [69]. Mechanistically, EGFR expression remains unchanged after treatment with EGCG but there is a reduction in its phosphorylated form [69].

#### 4.3.2. Chavibetol

Chavibetol in piper betel leaf acts as a bioactive molecule because it has anti-oxidant properties that protect against photosensitization-mediated lipid peroxidation in rat liver mitochondria [25,70]. Chavibetol also inhibits the formation of thiobarbituric acid reactive substances, recognised by-products of lipid degradation, and prevents the formation of lipid hydroperoxide following photo-irradiation of rat liver mitochondria [71]. Chavibetol is a known ROS scavenger and, therefore, appears to have an important role against photosensitization-induced biological damage. Further, chavibetol inhibits radiation-induced DNA strand breaks in a dose-dependent manner [72,73], a radio-protective effect that is mediated by its hydroxyl and superoxide radical scavenging properties.

#### 4.3.3. Flavonoids

Flavonoids include flavonols, flavones, isoflavones and anthocyanidins, amongst other compounds. They are present in foods of plant origin [74]. The potential of flavonoids to act as chemo-preventive agents has been described in many studies. Biochanin A and genistein (isoflavones) inhibit cell proliferation of OSCC cell lines through activation of Akt and MAP kinase signalling pathways [75,76] but their primary effect predominantly relates to their free radical scavenging activity. Silvan et al. [77] showed that the flavone apigenin reduced oxidative stress and modulated phase I and II detoxification cascades involved in xenobiotic biotransformation which, in turn, prevented oral tumour formation in Syrian hamsters. In addition, the administration of flavonoids to mice treated with radiation results in increased antioxidant enzyme activity and decreased lipid peroxidation by the scavenging of hydroxy radicals [78]. Reshma et al. [79] examined the effect of flavonoids on erythrocyte antioxidant defence potential in oral and oropharyngeal cancer patients and showed that erythrocytes from cancer patients responded to oxidative stress by increasing glutathione levels, whilst a decrease in glutathione levels was observed in the group receiving both radiotherapy and ocimum flavonoids.

#### 4.3.4. Tannins

For decades, it has been thought that tannins in *Areca catechu* are closely related to cancer development [80]. Later, it was shown [81] that tannin fractions derived from betel nut were genotoxic and caused chromatid breaks in mouse bone marrow cells. However, evidence has mounted that tannic acid, a specific form of tannin, possesses chemopreventive properties with possible uses in cancer prevention and as an adjuvant in cancer treatment [82]. Consistent with this view, tannic acid induces apoptosis, causes cell cycle arrest and limits the proliferation of cancer cells in animal models [83]. Further, Ta et al. [84] showed that low-dose tannic acid induced cell cycle arrest in the G2/M stage of the cell cycle in hypopharyngeal FaDu cells and when the dose of tannic acid was increased, apoptosis ensued with an increase in the cell population in G1. Mechanistically, tannic acid has been shown to inhibit the phosphorylation of the Jak2/STAT3 pathway in the YD-38 gingival OSCC cell line [85]. Hence, despite early studies, recent experimental evidence strongly suggests that tannins exert anti-cancer effects.

## 5. Fibroblast Senescence as a Possible Pathogenic Mechanism of OSMF and Progression to Malignancy

Various genetic and molecular mechanisms impact the development of OSMF and malignant transformation. Alterations occur in both oral keratinocytes and/or fibroblasts and lead to changes in the cell cycle, DNA and keratin formation, cell proliferation and survival, angiogenesis and fibrosis, epithelial-mesenchymal transition (EMT) and tissue hypoxia (reviewed in [86]). Although areca alkaloids are believed to be a major cause of neoplasia in individuals who habitually chew BQ containing areca nut, the exact mechanisms accounting for these putative carcinogenic effects in oral keratinocytes have not been elucidated. For example, areca nut extracts primarily produce carcinomas of the lung, liver and stomach in mice [1] and arecoline is known to be a weak mutagen in normal oral tissues [87]. Furthermore, in reports where arecoline has been described as both mutagenic and clastogenic, the studies have been undertaken using established cell lines where cell cycle checkpoints are disrupted [88]. It is possible, therefore, that other mechanisms involving the microenvironment play an important role in OSMF and tumour onset/progression.

### 5.1. Cellular Senescence Is Associated with OSMF

Early OSMF is characterised by epithelial atrophy, juxta-epithelial inflammation and the sub-epithelial connective tissues become avascular and thickened leading to collagen accumulation. In advanced disease, the fibrosis extends into the deeper tissues and is associated with an increased inflammatory cell infiltrate [89]. Many of these changes can be explained by senescence mechanisms. The atrophic epithelium can be attributed to senescence of epithelial cells and leads to the downregulation of basal stemness [90]. The reduced vascularity in OSMF may be attributable to senescence of endothelial cells, as suggested in a recent pathogenetic model [91]. Biologically active areca nut alkaloids stimulate fibroblasts to increase collagen synthesis and, concurrently, there is decreased activity of collagenases and reduced collagen degradation. These changes result in fibrosis of the underlying submucosa and are further associated with chronic inflammation and hypoxia [91].

Seminal work by Parkinson and colleagues has shown that OSMF is characterised by the presence of senescent fibroblasts in the subepithelial mesenchyme of OSMF patients [92,93]. Specifically, the frequency of senescent fibroblasts increases steadily from early OSMF through to advanced cases, with a further increase in advanced OSMF associated with dysplasia [93]. Therefore, there appears to be a progressive accumulation of senescent stromal cells in OSMF. Mechanistically, the same authors later showed that areca nut alkaloids induce both irreparable DNA damage and senescence in fibroblasts in vitro which creates a favourable environment for tumour progression [91]. Intrinsic oxidative damage was proposed as a potential cause of senescence in vitro and in vivo in OSMF, and, importantly, the use of anti-oxidants was capable of reducing the frequency of senescent cells [92]. Intriguingly, matrix metalloproteinases are overexpressed by senescent fibroblasts and may play an important role in the pathogenesis of the disease. Depleting the fibroblast population of senescent cells in OSMF drastically reduces the levels of MMP-1 and MMP-2 in conditioned medium suggesting that, as with cancer-associated fibroblasts (CAFs), there are both non-fibrogenic senescent fibroblasts within the cell population as well as the pro-fibrogenic cells that deposit collagen.

Overall, these data suggest that fibroblast senescence is central to the development of OSMF. However, do senescent fibroblasts promote malignancy?

### 5.2. Fibroblast Senescence Promotes Oxidative Stress and Oral Cancer Progression

We have shown that senescent fibroblasts share common characteristics with activated fibroblasts/myofibroblasts [94] and can drive OSCC progression via ROS and MMP production [95,96,97]. In particular, naturally occurring tumour-associated and normal fibroblasts that are artificially manipulated to become senescent are capable of promoting invasion of OSCC cells; this process is inhibited by antioxidants. Crucially, fibroblast senescence is induced by keratinocytes through ROS production in a TGF-β-dependent manner [96]. Further, we have shown that senescent fibroblasts secrete high levels of MMP-2 which cleaves cell-cell adhesion molecules and promotes invasion of malignant keratinocytes in vitro [97,98]. We have also shown that CAFs from genetically unstable OSCC (but not CAFs from genetically stable OSCC or normal fibroblasts) demonstrate high ROS production, can induce EMT and down-regulate a broad spectrum of cell adhesion molecules resulting in epithelial dis-cohesion and invasion [98]. Our results, therefore, show that an epithelial-stromal cross-talk exists that drives fibroblast senescence and this, in turn, modulates the malignant phenotype of keratinocytes. In view of the fact that fibroblast senescence in OSMF is independent of the continuous presence of areca nut and tobacco [92], it is possible that the disease-causing chemicals contained in BQ act indirectly, at least in part, by causing oxidative stress in keratinocytes. This, in turn, induces an irreversible senescent state in stromal fibroblasts which, ultimately, perpetuates OSMF and reinforces malignant transformation via induction of EMT (Figure 1).

### 5.3. A Pathogenic Model for OSMF and Possible Therapeutic Strategies

In our model, the main initiating event in the pathogenesis of OSMF is the production of ROS by the BQ constituents. This, in turn, leads to oxidative stress and senescence in keratinocytes, fibroblasts and endothelial cells. Senescence in keratinocytes induces epithelial atrophy, whilst senescence in endothelial cells accounts for a decrease in vascularity and leads to the development of a hypoxic state. In fibroblasts, the mechanism is associated with irreparable DNA damage and development of the SASP. There is then a positive feedback because the SASP reinforces ROS production and the cycle is repeated and amplified (Figure 1). Finally, the inflammatory response enhances fibrogenesis [99]. Together, these key drivers of OSMF form the milieu that promotes its malignant transformation. In summary, our model points to the involvement of senescence, ROS and EMT in the pathogenesis of OSMF and its progression to OSCC.

This model would be amenable to several preventive and therapeutic strategies for OSMF including the use of senostatics, senolytics and anti-oxidants. Inhibiting SASP production/function (senostatics) has been the goal of past research from an early stage, but recent data have shown that the SASP is cell-type specific making this approach less practical. Some potential drug targets, particularly those associated with inhibition of the inflammasome and plasminogen activator inhibitor-1 [100], can also regulate the SASP. For example, the MAPK14 inhibitors and steroids, but not other kinase inhibitors, have been reported to inhibit the secretion of a diverse array of SASP proteins [101].

Senolytics are a class of drugs that selectively target senescent cells for destruction. These drugs include BCL-XL inhibitors, finestin, ouabain, quecertin and dasatinib [102]. Senolytics ameliorate the effects of many age-related diseases in mice and clinical trials have begun to test their effects in human disease. Although there is some evidence that senolytics can ameliorate fibrosis in humans [103], these drugs can only be administered for a short time due to known side effects (thrombocytopenia). 

Drugs or natural compounds with antioxidant properties such as aloe vera, curcumin and lycopene have been used extensively in the treatment of OSMF, unfortunately with inconsistent results [104]. A recent network meta-analysis has proposed that herbal derivatives may be more effective in the management of mouth opening in OSMF,, particularly if used at an early stage [105]. However, one of the issues with the efficacy of antioxidant compounds is that senescence is an irreversible state that continues even if the stimulus that caused it ceases. Therefore, whilst ROS are instrumental in the generation of senescence in the early stages of OSMF, the process is likely to continue independent of the presence of anti-oxidants. Prevention, therefore, is key to reduce the burden of OSMF.

## 6. Conclusions

In patients with OSMF, progression to cancer is thought to be related to smoking and alcohol consumption [106] but, also, current thinking indicates that fibrosis-specific mechanisms are active. This review describes the possible role of potentially carcinogenic constituents of betel quid, predominantly alkaloids and polyphenols. The alkaloids and their derivatives are found in varying concentrations and proportions within the areca nut and, in the saliva of betel quid chewers. Whilst polyphenols such as tannins and flavonoids have anti-oxidant effects and could serve as chemopreventive agents, under certain environmental conditions they have the potential to enhance the genotoxic and carcinogenic effects of the alkaloids, with specific implications for the development of OSCC.

Despite its known potential to cause disease, betel quid chewing plays an important role in religious practices, cultural rituals and social customs throughout Asia and the Pacific rim. The eradication of this habit, therefore, may not be realistically achievable. Rather than impose a change in culture and way of life, it might be possible to identify mixtures that have a greater or lesser risk of OSMF, and, therefore,, by modulating fibrosis through betel quid regulation, malignant transformation in OSMF may be reduced together with the morbidity of OSMF.

Our unpublished data suggest that chewing betel quid mixtures with the addition of betel inflorescence (BI) increases the risk of OSMF. Conversely, the use of betel leaf in the mixture is likely to be protective. For all of these reasons, we believe that a better understanding of the effects of each BQ ingredient would help develop a sustainable chemopreventive strategy for this disease. The present review highlights our lack of knowledge in the field and the urgent need for further research. In the future, therefore, we suggest that further work should involve a broad range of geographic regions where different betel quid constituents are consumed. We believe that the use of consistent and contemporary methodology to measure carcinogenic risk is essential. In this way, we would be able to address the impact of changes in the concentrations of different constituents and the consequences of areca nut maturity and processing.

Based on our own findings and those of others, we propose a pathogenic loop involving a cross-talk between keratinocytes and fibroblasts that fuels the development of OSMF and, possibly, its malignant transformation. Specifically, the consequence of cellular senescence is the generation of the SASP, the production of ROS and the induction of DNA double strand breaks in keratinocytes. Only when the keratinocytes escape from senescence does malignant transformation ensue, but the myofibroblasts persist because they evade immune clearance. We, therefore, suggest the use of antioxidants and senolytics as novel mechanism-based treatment modalities for OSMF.

## Figures and Tables

**Figure 1 ijms-23-01637-f001:**
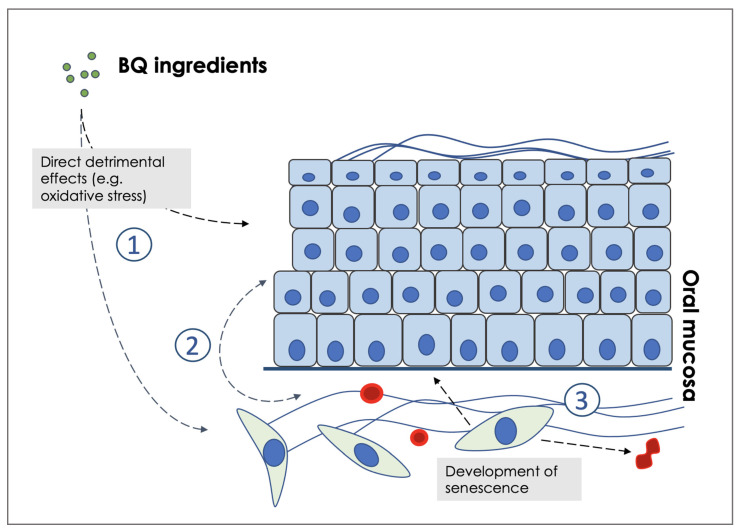
Proposed pathogenic model for oral submucous fibrosis. Briefly, BQ constituents induce direct damage to both keratinocytes and fibroblasts resulting in DNA damage and production of reactive oxygen species (1). There is cross-talk between epithelial and mesenchymal compartments that amplifies the pathogenic loop and leads to fibrosis and hypoxia (2). Senescent fibroblasts promote the development of a microenvironment permissive for cancer progression (3).

**Table 1 ijms-23-01637-t001:** Heat map of the various chemical constituents found in betel quid without tobacco according to representative articles that met the eligibility criteria.

	Jain et al., 2017 [24]	Franke et al., 2015 [25]	Cao et al., 2019 [26]	Srimany et al., 2016 [27]	Shirname et al., 1983 [28]	Kadi et al., 2013 [29]	Tang et al., 2017 [30]	Lord et al., 2002 [31]	Adhikari et al., 2015 [32]
Arecoline									
Arecaidine									
Guvacoline									
Guvacine									
Isoguvacine									
Acatechu A									
Acatechu B									
Homoarecoline									
Arecatemine A									
Arecatemine B									
Arecatemine C									
Flavonoids									
Tannins									
Carbohydrates									
Fatty acids									
Chavibetol

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
