# Peer review of "Molecular Mechanisms of Malignant Transformation of Oral Submucous Fibrosis by Different Betel Quid Constituents—Does Fibroblast Senescence Play a Role?"

_ijms, 2022, doi:10.3390/ijms23031637_

Round 1

Reviewer 1 Report

This review article describes an precancerous disorder and points out the importance of fibroblast cells in the pathogenic process. The theme is not new and recently other authors have made also similar work but with more clinical aspects such as Sonia Gupta and Manveen Kaur Jawanda that by the way the present authors must also cite in this paper. However, this is an important paper that brings new developments in the area of BQ componentes, especially for Asian people, a target population. 

There are some comments that authors must address.

In the introduction, please correct the sentence regarding the % of malignant transformation of OSMF. I read the paper and authors refers % between 1,5 and 15%. Where did you find the 30% value?

Do you have % of mortality in the literature? Did you search for survival articles? Please clarify and insert the information.

In section 5, the authors refer the possible effects in the keratin. Can you please clarify these effects in Keratin specifically?

In the conclusion I believe that is more accurate refer: “This review describes…” instead of “this review investigated”.   

Please insert the reference of Sonia Gupta (2021):

“Oral submucous fibrosis: An overview of a challenging entity”

Author Response

Please see our response to the criticisms raised. thanks, Nicola

REVIEWER #1

This review article describes an precancerous disorder and points out the importance of fibroblast cells in the pathogenic process. The theme is not new and recently other authors have made also similar work but with more clinical aspects such as Sonia Gupta and Manveen Kaur Jawanda that by the way the present authors must also cite in this paper. However, this is an important paper that brings new developments in the area of BQ componentes, especially for Asian people, a target population. 

There are some comments that authors must address.

In the introduction, please correct the sentence regarding the % of malignant transformation of OSMF. I read the paper and authors refers % between 1,5 and 15%. Where did you find the 30% value?Do you have % of mortality in the literature? Did you search for survival articles? Please clarify and insert the information.

Thanks for this valuable comment. As rates malignant transformation differ between studies, we have provided two references to support the data. With regard to mortality, it is generally related to transformation to OSCC. Accordingly, we have redrafted the sentence as follows:

“OSMF is a major global health issue because of its high malignant transformation rate (up to 15-30% according to different studies) [4, 5] and subsequent high mortality due to OSCC [5].”

In section 5, the authors refer the possible effects in the keratin. Can you please clarify these effects in Keratin specifically?

Thanks for this comment. In par.5, we have referred to keratinocytes extensively, and highlighted their role in the pathogenesis of submucous fibrosis. Keratin was only mentioned once, at the beginning of the paragraph, to give a roundup of the many changes that have been reported. We then focused on oxidative stress mechanisms as the initial pathogenic event.

Keratins have been correlated to OSMF severity (Lalli et al., JOPM 2008) and loricrin – a major component of the cornified envelope keratins – has been used as a diagnostic marker of OSMF (Mahapatra et al., JPBS 2020). However, this was beyond the scope of this review.

In the conclusion I believe that is more accurate refer: “This review describes…” instead of “this review investigated”.   

Many thanks for this comment, we have amended the manuscript accordingly. It now reads “This review describes the possible role of potentially carcinogenic constituents of betel quid”

Please insert the reference of Sonia Gupta (2021): “Oral submucous fibrosis: An overview of a challenging entity”

We appreciate this suggestion, and have inserted this reference at the outset of our review, as follows: “The historical background, etiological factors, pathogenesis, clinical features, differential diagnosis, and management of this condition have been reviewed recently [6].”

Reviewer 2 Report

The publication by Zhang and colleagues is interesting but requires some explanation.

Polyphenols are rather neutral at low concentrations for normal cells. Could the polyphenol concentrations in BQ play a role in the development of OSCC? The authors indicate that polyphenols may be one of the causative factors in the initiation of carcinogenesis in BQ chews, so they should provide more evidence that these compounds may play a role in OSMF and OSCC development in the oral microenvironment and in the presence of a specific mixture of BQ components.

Is there information in the literature on the relationship between smoking or drinking alcohol and BQ chewing, regardless of tobacco in BQ, and the incidence of OSMF and OSCC.

The authors should include a scheme of the potential carcinogenic mechanism of BQ components.

Author Response

Please see our response to the criticisms raised. thanks, Nicola

REVIEWER #2

The publication by Zhang and colleagues is interesting but requires some explanation.

Polyphenols are rather neutral at low concentrations for normal cells. Could the polyphenol concentrations in BQ play a role in the development of OSCC? The authors indicate that polyphenols may be one of the causative factors in the initiation of carcinogenesis in BQ chews, so they should provide more evidence that these compounds may play a role in OSMF and OSCC development in the oral microenvironment and in the presence of a specific mixture of BQ components.

Many thanks for this important comment, we appreciate that this information is key for a correct understanding of the role of polyphenols in the development of OSMF.

In the main, we provided data to support their anti-oxidant (hence protective) capacity (e.g. line 306 onwards “Polyphenols are commonly known to exert antioxidant effects but can also act as pro-oxidants to induce apoptosis [63]” and line 317 onwards “In summary, the majority of the effects attributed to these compounds are linked to their antioxidant and anti‑inflammatory properties…”. Data related to the potential pro-oxidative or dual effects of polyphenols and their possible interaction with other BQ components and the oral environment were also provided (refs 8, 46, 61, 63, 66).

To make it clear to the reader that polyphenols are mainly anti-oxidants, we summarised their role at the beginning of the paragraph, as follows: “In the main, polyphenols are regarded as anti-oxidants and as such would play a protective role  in OSMF. However, their biological properties can be significantly modified by the chemical environment. For example, it is known….

Is there information in the literature on the relationship between smoking or drinking alcohol and BQ chewing, regardless of tobacco in BQ, and the incidence of OSMF and OSCC.

Many thanks for this comment. BQ chewing seems to be the only significant risk factor for the development of OSMF (Ariyawardana et al., JOPM 2006). We do understand, however, that progression to cancer may involve the same factors as non-fibrosis related OSCC (e.g. alcohol, Chuang et al., Oral Oncol 2018), and have added the following statements to make it clear:

Line 211: “Smoking and drinking alcohol are well-known risk factors of OSCC, however malignant transformation of OSMF involves additional and potentially different pathways that are specific to OSMF. For example…”

and at the beginning of Conclusion: “In patients with oral submucous fibrosis, progression to cancer may be related to smoking and alcohol [106], but also involves fibrosis-specific mechanisms. This review describes…

The authors should include a scheme of the potential carcinogenic mechanism of BQ components.

We have summarised our model in a schematic picture (Figure 1).